# Data Selection Curriculum for Abstractive Text Summarization

**Shichao Sun[1], Ruifeng Yuan[1], Jianfei He[2], Ziqiang Cao[3], Wenjie Li[1], Xiaohua Jia[2]**
[1]The Hong Kong Polytechnic University, [2]City University of Hong Kong, [3]Soochow University
{csssun,csryuan}@comp.polyu.edu.hk, jianfeihe2-c@my.cityu.edu.hk
zqcao@suda.edu.cn, cswjli@comp.polyu.edu.hk, csjia@cityu.edu.hk

## Abstract

Abstractive Text Summarization (ATS) models are commonly trained using large-scale data that is randomly shuffled. However, the impact of data selection and data ordering on ATS models remains a relatively unexplored research area, where a significant challenge lies in accurately assessing the learning difficulty of each training instance. This study introduces a **D**ata **S**election **C**urriculum (**DSC**) scoring system that incorporates both the difficulty of improving ATS model via an instance and the expected performance on this instance. By selectively excluding excessively simple and overly complex instances, the training efficiency can be optimized. Furthermore, curriculum learning is integrated to accelerate convergence and improve performance by gradually increasing the learning difficulty, inspired by human learners. Experimental results on the CNN/DailyMail dataset demonstrate that our approach surpasses potent baselines, utilizing a mere 20% of the available instances.

## 1 Introduction

Abstractive Text Summarization (ATS) aims to generate concise summaries while preserving essential content. Recent studies (Liu et al., 2023b; Zhang et al., 2023; Liu et al., 2023a) have shown that Large Language Models (LLMs), including GPT-3.5 (Ouyang et al., 2022), can produce summaries more favored by human annotators compared to reference summaries from well-established datasets, such as CNN/DailyMail (Hermann et al., 2015). Additionally, Liu et al. (2023a) experimentally assert that contrastive learning based methods applied to smaller summarization models, like BART (Lewis et al., 2020), can deliver performance on par with LLMs. This finding renews the importance of training smaller models via contrastive learning, as it offers the advantage of reducing computational costs. However, ATS models are commonly trained using large-scale data that is randomly shuffled.

This study aims to explore the potential for optimizing ATS models through the strategic utilization of data selection and curriculum learning (Bengio et al., 2009) in conjunction with contrastive learning techniques. It is important to note that not all data is equally valuable, and the presence of redundant or even detrimental examples can impede the performance of ATS systems (Mohiuddin et al., 2022). Furthermore, the ordering in which data is presented during model training, as emphasized by curriculum learning principles, can have a significant impact on both the efficiency and effectiveness of the learning process. Consequently, there is a fundamental necessity to develop a scoring system that can accurately assess the learning difficulty of individual samples and furtherly can be used for data selection and curriculum learning.

In this study, we introduce a novel scoring system called the **D**ata **S**election **C**urriculum (**DSC**) score, which serves as a measure of learning difficulty. The DSC score considers two essential factors: the difficulty of enhancing the ATS model through a specific instance and the expected performance of the model on that particular instance. Instances on which the current model already exhibits promising performance and stands to benefit significantly from slight adjustments are assigned as a lower learning difficulty. This is because these instances have the potential to contribute more to the final performance. As for the first factor, we propose a novel **M**argin-aware **L**ist-wise **R**anking (**MLR**) loss, which introduces a dynamic margin instead of constant margin, which the state-of-the-art BRIO (Liu et al., 2022) uses in the list-wise ranking loss for contrastive learning. It can better reflect the fine-grained misalignment between sequential likelihood and performance. Through the DSC score, we can effectively select and prioritize instances during the training process, focusing on those that offer the greatest opportunities for enhancing the ATS model's overall performance.

We empirically validated our method on the CNN/DailyMail dataset. The results suggest that our method surpasses several robust baselines using fewer than 20% of instances. Additionally, experimental findings highlight that our MLR loss is superior to the BRIO loss.

## 2 Method

### 2.1 Preliminary

As evidenced in recent research, contrastive learning has the capacity to achieve superior performance in text summarization (Liu et al., 2022; Zhang et al., 2022; Zhao et al., 2023b; Liu et al., 2023a; Zhao et al., 2023a). These methodologies adopt a novel three-stage training protocol, comprising pre-training, fine-tuning, and calibrating stages. The calibrating stage employs diverse beam search (Vijayakumar et al., 2018) to generate a variety of candidates, which are subsequently ranked based on their metric scores. A ranking loss tunes the parameters, aligning the sequence likelihood with the established order. List-wise rank loss $\mathcal{L}_{cal}$ from BRIO (Liu et al., 2022), along with cross-entropy loss $\mathcal{L}_{xent}$, has been shown to yield promising results (Liu et al., 2022; Zhang et al., 2022; Zhao et al., 2023b; Liu et al., 2023a). The overall loss $\mathcal{L}$ is delineated as follows:

$$\mathcal{L} = \mathcal{L}_{xent} + \gamma \mathcal{L}_{cal}$$

$$\mathcal{L}_{xent} = -\frac{1}{|Y|} \sum_{t=1}^{|Y|} \log p_\theta(y_t | X, y_0, \cdots, y_{t-1})$$

$$\mathcal{L}_{cal} = \sum_i \sum_{i<j} max(0, \lambda_{ij} + s(Y_j^{'}) - s(Y_i^{'}))$$

$$\lambda_{ij} = \lambda \times (j - i)$$

$$s(Y^{'}) = \frac{\sum_{t=1}^{|Y^{'}|} \log p_\theta(y_t^{'} | X, y_0^{'}, \cdots, y_{t-1}^{'})}{|Y^{'}|^\alpha}$$

$$(1)$$

where $X, Y, Y^{'}$ respectively denotes the input document, the ground truth summary, and candidate summaries from diverse beam search. The (candidate or ground truth) summary $Y$ consists of $|Y|$ tokens, i.e., $y_0, y_1, \cdots, y_{|Y|}$, and $|Y|$ is the length of the summary $Y$. $y_0$ is the pre-defined start token. $\mathcal{L}_{xent}$ is the mean of the negative log-likelihood of all ground truth summary tokens. $Y_j^{'}$ and $Y_i^{'}$ are two different candidate summaries and $i, j$ are their rank index in the ranked list. It means that the quality of $Y_i^{'}$ is better than $Y_j^{'}$ in terms of the

pre-defined metric score when $i < j$. $\lambda$ is the margin between two adjacent candidates. The score $s(Y^{'})$ considers normalizing the candidate's length by a penalty hyperparameter $\alpha$. $p_\theta$ denotes the fine-tuned language model with parameter $\theta$, like BART. $\gamma$ is the weight of the calibration loss, which prevents models from deviating significantly from their fine-tuned cross-entropy objective.

### 2.2 MLR Loss

The calibration loss $\mathcal{L}_{cal}$ previously discussed uses a constant margin, denoted as $\lambda$, for every pair of adjacent candidates. The utilization of a constant margin may be counter-intuitive since this method may demonstrate limitations in certain scenarios, such as cases where different pairs of adjacent candidates exhibit varying differences in metric scores. To rectify this, we propose a novel **M**argin-aware **L**ist-wise **R**anking (**MLR**) loss that can serve as a more flexible replacement for $\mathcal{L}_{cal}$. The MLR loss is defined as follows:

$$\mathcal{L}_{mlr} = \sum_i \sum_{i<j} max(0, r_{ij} + s(Y_j^{'}) - s(Y_i^{'}))$$

$$r_{ij} = \beta \times (m(Y_i^{'}) - m(Y_j^{'}))$$

$$(2)$$

where $m(*)$ denotes the metric function, which could be BERTScore (Zhang et al., 2020) or ROUGE (Lin, 2004), and $\beta$ is a hyperparameter responsible for scaling the difference between two candidates. Please note that the value of $m(*)$ is within the range of 0 to 1, with a higher $m(*)$ score indicating superior performance. Consequently, this setup makes the calibration loss aware of the margin in metric scores, enabling more precise calibration training. In essence, the $\mathcal{L}_{mlr}$ loss can better define the difficulty faced by the model in transitioning to a more enhanced version of parameters, unlike the $\mathcal{L}_{cal}$, which fails to facilitate fine-grained changes due to its constant margin.

### 2.3 Difficulty Metric

The difficulty metric, represented as $\mathcal{D}$, incorporates both the MLR loss and expected model performance. It is defined as follows:

$$w_i = \frac{\exp(s(Y_i^{'}))}{\sum_j \exp(s(Y_j^{'}))}$$

$$\mathcal{E} = \sum_i m(Y_i^{'}) w_i$$

$$(3)$$

$$\mathcal{D} = 1 - \mathcal{E} + \mathcal{L}_{mlr}$$

where the expected model performance $\mathcal{E}$ is defined as the expected metric scores of all candidates that the model generates by diverse beam search.

## 2.4 Data Selection Curriculum

We rank training data based on the difficulty metric discussed above. Inspired by the success of Mohiuddin et al. (2022), we discard data instances that are too easy or too hard to learn. Assuming a Gaussian Distribution of the training data, the data selection window can be defined as $[\mu(\mathcal{D}) - \delta\sigma(\mathcal{D}), \mu(\mathcal{D}) + \delta\sigma(\mathcal{D})]$, where $\mu(\mathcal{D}), \sigma(\mathcal{D})$ represent the mean and standard deviation respectively, and $\delta$ is a hyperparameter that is used to adjust the window size of data selection.

## 3 Experiment

### 3.1 Implementation Details

We implement our method on top of the BRIO open-source codebase [1] and adhere to most of its experimental settings as defined by Liu et al. (2022). Differences in settings and the reasons behind these will be discussed in this section. As our approach is model-agnostic, we initialize it using the BRIO checkpoint [2] to ensure quick convergence. To better understand the data effect, we use a constant learning rate of 1e-6 instead of a learning rate schedule, thus reducing the optimizer's impact. Liu et al. (2022) found that the use of ROUGE scores in the metric function $m(*)$ (referenced in Equations (2) and (3)) resulted in better performance than BERTScore. Therefore, we follow this approach, using the mean value of ROUGE-1, ROUGE-2, and ROUGE-L as $m(*)$ to ensure a range from 0 to 1, the same as implemented in BRIO. $\beta$ is set to 0.1 for all experiments.

### 3.2 Data

We conduct experiments on the CNN/DailyMail dataset[3] (Hermann et al., 2015), a large-scale news dataset with 287K/13K/11K instances for training/validation/testing respectively. As per previous work (Nallapati et al., 2016; See et al., 2017; Liu et al., 2022), we use the news articles (averaging 791.6 words) as the source documents and the associated highlights (averaging 3.75 sentences or 55.6 words) as summaries.

---

[1] https://github.com/yixin17/brio
[2] https://huggingface.co/Yale-LILY/brio-cnndm-uncased
[3] https://cs.nyu.edu/~kcho/DMQA/

In this study, we conducted an analysis of the training instances using the difficulty metric discussed earlier (in Section 2.3). The results of this analysis are presented in Figure 1. We observed that the distribution of the difficulty scores closely approximates a Gaussian Distribution. The observation provides strong support for our hypothesis in Section 2.4. This verification indicates that our DSC method can be effective in identifying and prioritizing training instances with the mean and standard deviation.

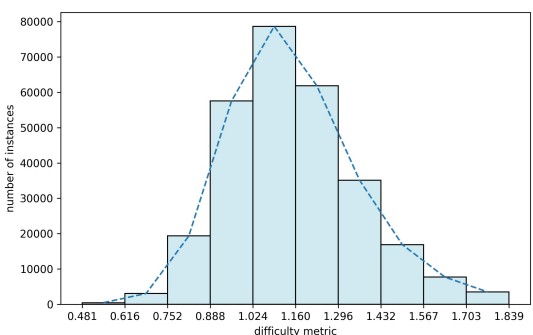

Figure 1: Difficulty metric histogram.

### 3.3 Baseline

For comparison, we choose several impressive models as baselines. **BART** (Lewis et al., 2020), a large pre-trained Seq2Seq model, is fine-tuned on CNN/DailyMail. **BRIO** (Liu et al., 2022) is a promising model based on contrastive learning, which is initialized by BART. **BRIO-Loop** is an iterative version of BRIO, trained on the candidates generated by BRIO. **MLR** substitutes for the calibration loss of BRIO-Loop. Finally, **DSC** represents our proposed method which uses data selection curriculum along with the MLR loss.

## 4 Results

### 4.1 Overall Evaluation

Table 1 shows the evaluation results of various methods on CNN/DailyMail. Notably, our proposed DSC method outperforms the strong baseline models in terms of ROUGE scores and BERTScore. This improvement demonstrates the effectiveness of the DSC method in enhancing the abstractive text summarization task. Moreover, MLR achieves higher ROUGE scores and BERTScore than BRIO-Loop. This finding underscores the effectiveness of dynamic margin calibration in sequential likelihood calibration, as opposed to a constant margin.

| Method | R-1 | R-2 | R-L | BS |
|---|---|---|---|---|
| BART* | 44.29 | 21.17 | 41.09 | 27.38 |
| BRIO* | 47.78 | 23.55 | 44.57 | 32.11 |
| BRIO-Loop* | 48.01 | 23.80 | 44.67 | – |
| MLR | 48.17$^\dagger$ | 23.95$^\dagger$ | 44.92$^\dagger$ | 35.25$^\dagger$ |
| DSC | **48.62$^\dagger$** | **24.14$^\dagger$** | **45.31$^\dagger$** | **35.90$^\dagger$** |

Table 1: Overall evaluation results. $^\dagger$: significantly better than the baseline model ($p < 0.01$). *: results reported in the original papers. R-1/2/L are ROUGE-1/2/L $F_1$ scores. BS denotes BERTScore.

## 4.2 Coefficients of the Data Selection

The hyperparameter $\delta$ can control the size of the data selection window. A larger $\delta$ results in a larger window for selected data, including more easy and hard instances for training. We explore the impact of different hyperparameter $\delta$ values on model performance. Our experiments, reported in Table 2, reveal that including more data is not always beneficial. Performance declines with larger $\delta$ values. Conversely, too few instances also lead to poor performance when $\delta$ is small.

| $\delta$ | R-1 | R-2 | R-L | BS |
|---|---|---|---|---|
| 0.50 | 48.26 | 23.92 | 45.00 | 35.42 |
| 0.75 | 48.44 | 24.09 | 45.15 | 35.73 |
| 1.00 | 48.46 | 24.11 | 45.22 | 35.75 |
| 1.25 | **48.62** | **24.14** | **45.31** | **35.90** |
| 1.50 | 48.59 | 24.14 | 45.24 | 35.71 |
| 1.75 | 48.51 | 24.13 | 45.24 | 35.64 |
| 2.00 | 48.41 | 24.12 | 45.22 | 35.59 |

Table 2: Model performance with different $\delta$ coefficients. R-1/2/L are ROUGE-1/2/L $F_1$ scores. BS denotes BERTScore.

## 4.3 Selected Data Ratio

To investigate the impact of the number of selected instances, we conducted experiments using varying ratios of selected data from the dataset. The model was trained on the selected data for one epoch, and without early stopping, the latest checkpoint was used for testing. Alternatively, with early stopping, the checkpoint with the highest ROUGE scores was selected. The results, depicted in Figure 2, indicate that the quantity of data does not necessarily correlate with improved performance. Surprisingly, an excessive number of instances may even lead to

a degradation in performance. Moreover, we observed that the training process tends to favor easier instances, as optimal performance was achieved with less than 20% of the available instances.

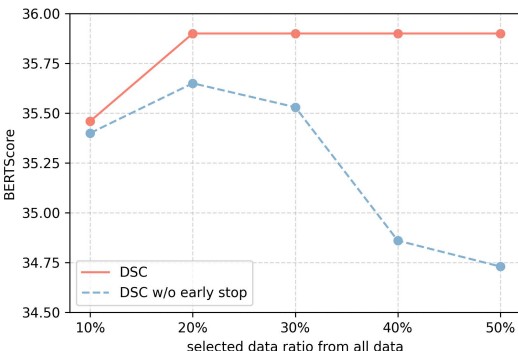

Figure 2: DSC performance with different data ratios.

We validate our methods on XSum (Narayan et al., 2018) from the same start point of BRIO. We initialize our model using "google/pegasus-xsum"[4] and use the selected 20% instances to train. The BERTScore of our model (52.88) beats BRIO (41.08) by a large margin, so the summary from our model is more similar to the reference than BRIO.

## 4.4 Ablation Study

| Method | R-1 | R-2 | R-L | BS |
|---|---|---|---|---|
| BRIO* | 47.78 | 23.55 | 44.57 | 32.11 |
| + MLR | 48.17$^\dagger$ | 23.95$^\dagger$ | 44.92$^\dagger$ | 35.25$^\dagger$ |
| + MLR DS | 48.36$^\dagger$ | 24.06$^\dagger$ | 45.04$^\dagger$ | 35.54$^\dagger$ |
| + MLR DS CL | **48.62$^\dagger$** | **24.14$^\dagger$** | **45.31$^\dagger$** | **35.90$^\dagger$** |

Table 3: Ablation Study. $^\dagger$: significantly better than the baseline model ($p < 0.01$). *: results reported in the original papers. R-1/2/L are ROUGE-1/2/L $F_1$ scores. BS denotes BERTScore.

We also conducted separate experiments to evaluate each sub-model. **MLR** represents our novel contrastive ranking loss. **DS** is shorthand for data selection, which refers to training the model using selected data instead of all instances. **CL** denotes curriculum learning, which trains the model in the from-easy-to-hard ordering. Table 3 shows that all methods we proposed contribute to the improvement of summarization performance.

---

[4]https://huggingface.co/google/pegasus-xsum

## 5   Conclusion

This paper investigates the impact of data on text summarization. We propose a learning difficulty metric for data selection and curriculum learning. This score incorporates a Margin-aware List-wise Ranking (MLR) loss, further enhancing contrastive learning due to its dynamic margin. Experimental results indicate that our data selection curriculum can outperform strong baselines using fewer than 20% instances from the CNN/DailyMail dataset.

## 6   Limitations

We acknowledge that the conclusions presented in the paper are derived from analyzing a single dataset specifically focused on English news articles. While this dataset is widely accepted as a benchmark for summarization tasks, it is important to recognize that the results obtained may not be applicable to other languages or domains. Therefore, caution should be exercised when generalizing these findings beyond the scope of the dataset. Despite the limitation, the findings of this study are valuable in terms of providing insights into the potential enhancements that can be achieved in abstractive summarization by employing strategic data selection and curriculum learning techniques. These insights contribute to the ongoing exploration of improving summarization methods.

## Acknowledgements

The work described in this paper was supported by Research Grants Council of Hong Kong (PolyU/15203617 and PolyU/5210919), National Natural Science Foundation of China (61672445, 62076212, 62106165).

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
