# OpenReview forum: "Data Selection Curriculum for Abstractive Text Summarization"
_EMNLP/2023/Conference — EMNLP 2023 Findings_

### Official Review · Reviewer_pCuk · 2023-07-31

**Soundness:** 3

**Excitement:**

3: Ambivalent: It has merits (e.g., it reports state-of-the-art results, the idea is nice), but there are key weaknesses (e.g., it describes incremental work), and it can significantly benefit from another round of revision. However, I won't object to accepting it if my co-reviewers champion it.

**Paper Topic And Main Contributions:**

In this paper, they design a Data Selection Curriculum for Abstractive Text Summarization. They propose a data selection method for the contrastive learning based approach.

1.  they introduce a dynamic margin instead of a constant margin to improve the loss function proposed in BRIO, called MLR loss.

2. they use both MLR and expected model performance to design a difficulty metric.

3. they use difficulty scores to select data for training.

**Questions For The Authors:**

Q1: β is a key parameter. Are there any experiments that suggest it should take a value of 0.1?

Q2: In Figure 1, the authors show the difficulty metric of all training data,  I would like to know how the metric is calculated. And what are the parameters \theta of the model?

Q3: CL trains the models in order from easiest to hardest. I know the process of sorting the data, but not how they add the data to the training set.

Q4: In Eq(3), the second sub-equation, I wonder what is the sum operation.


**Reasons To Accept:**

1. See the above part.

2. The motivation is clearly described. They combine the loss of model to evaluate the difficulty of the instance, it's reasonable.

3. Experimental results demonstrate the method proposed in this paper is effective.

It is well-written and easy to follow.

**Reasons To Reject:**

1. CL strategy proposed in this paper is model-agnostic, but it is designed for contrastive learning models, not for the Abstractive Text Summarization task.

2. The design of their difficulty measure is not clearly explained, i.e., why the formula in section 2.3 is designed the way it is; the difficulty is a very important part of curriculum learning, and readers would be more likely to understand it if there were a more out explanation of it.

3. There are some details that are not clearly described, see Questions For The Authors.

**Reproducibility:**

4: Could mostly reproduce the results, but there may be some variation because of sample variance or minor variations in their interpretation of the protocol or method.

**Reviewer Confidence:**

4: Quite sure. I tried to check the important points carefully. It's unlikely, though conceivable, that I missed something that should affect my ratings.

---

> ### Author Rebuttal · Authors · 2023-08-29
>
> We are extremely grateful for the positive assessment and highly detailed and constructive feedback on our paper.
>
> **Weakness:** Explanation why the difficulty measure is designed as mentioned in section 2.3.
>
> We design a difficulty measure metric for an instance by considering the expected performance ($\mathcal{E}$) of the current model in this instance and the effort of transitioning the current model to a stronger version.
>
> If the current model can obtain the promising performance for this instance, it means that the current model ''knows'' this instance well and the current model can easily learn this instance. We consider the generation process as sampling so we can obtain some sampled summaries $Y$ and their probability-based scores $s(Y)$.  We can define the expected performance as the weighted sum of the summary metric score $m(Y)$. The weight is the softmax scores of probability-based scores $s(Y)$. A larger expected score ($\mathcal{E}$) means easier to learn, so we use   (1-$\mathcal{E}$) to represent the difficulty. The score  (1-$\mathcal{E}$) is large so the instance is hard to learn.
>
> As mentioned in section 2.2, mlr loss can reflect the effort that the current model needs to take for achieving stronger performance.
>
> Therefore, it is intuitive to define this difficulty measure score as the combination of the above two scores.
>
>
>
> **Q1:** β is a key parameter to balance the order of magnitude between the model-predicted scores and the reference-based scores (like ROUGE or BERTScore). If this value is too small, it means that the margin between different summaries is similar (very small, like 1e-6 and 2e-6), and this method may take a slight effect. We obtain this value 0.1 on CNNDM by observing some model-predicted scores and their ROUGE scores. To further validate the effect of this parameter, we conduct experiments on XSum. We found the β is set to 1, we can obtain the best ROUGE performance 49.27/25.75/40.40 against 49.01/25.41/40.18 when β is set to 0.01.
>
>
>
> **Q2:** The difficulty score in Figure 1 is calculated as mentioned in section 2.3. And the model parameter is our initialized model ''Yale-LILY/brio-cnndm-uncased'' from huggingface.
>
>
>
> **Q3:** We calculate the difficulty scores as mentioned in section 2.3, and then we use this score to sort all training instances. Please note that the data distribution closely approximates a Gaussian Distribution as shown in Fig 1, so we can use the mean value and standard deviation to find suitable data. We test different $\delta$ values in Table 2 and select the best one.
>
>
>
> **Q4:** As mentioned in the response of weakness, the second sub-equation is defined as the expected value of the sampling summaries score (ROUGE or BERTScore) with their probability-based score. After softmax, the weight can be viewed as the probability of generating each summary from the model. So, this weighted sum can be viewed as the expected summaries score. It can reflect the average performance of the current model with sampling.

---

### Official Review · Reviewer_r3Wo · 2023-08-03

**Soundness:** 3

**Excitement:**

3: Ambivalent: It has merits (e.g., it reports state-of-the-art results, the idea is nice), but there are key weaknesses (e.g., it describes incremental work), and it can significantly benefit from another round of revision. However, I won't object to accepting it if my co-reviewers champion it.

**Paper Topic And Main Contributions:**

In this work, the authors investigate the impact of data on text summarization; they proposed a learning difficulty metric for data selection and curriculum learning. Due to its dynamic margin, this score incorporates a margin-aware list-wise ranking loss to enhance contrastive learning. By selectively excluding straightforward and overly complex instances, the CNN/DailyMail dataset get better results.


**Reasons To Accept:**

1.	This paper is easy to read. Although the length of this paper is small, it provides the necessary background and explanation.
2.	Based on BRIO model, the authors change the margin between two adjacent candidates of contrastive loss from a fixed distance to a dynamic metric-based calculation.


**Reasons To Reject:**

1.	The author has only conducted experiments on CNN/DailyMail dataset and obtained good results. Can it be generalized to other datasets?
2.	The overall workload is less. The authors should provide the result on the XSum dataset or other datasets. This needs to be verified.
3.	On the CNN/DailyMail dataset, the authors utilized a data selection curriculum that can outperform baselines using 20% instances on the CNN/DailyMail dataset. Is it possible that this improvement is caused by selecting the most favorable part of the model data for training?


**Reproducibility:**

4: Could mostly reproduce the results, but there may be some variation because of sample variance or minor variations in their interpretation of the protocol or method.

**Reviewer Confidence:**

4: Quite sure. I tried to check the important points carefully. It's unlikely, though conceivable, that I missed something that should affect my ratings.

---

> ### Author Rebuttal · Authors · 2023-08-29
>
> Thank you very much for your detailed comments.
>
> **Weakness 1 & 2:**
>
>
> We try to repeat our experiments on XSum from the same start point of BRIO. We initialize our model using "google/pegasus-xsum" using the selected 20% instances to train. The results are in the following table. The performance of ROUGE-2 and ROUGE-L are competitive. Besides, the BERTScore of our model beats BRIO, so the summary from our model is more similar to the reference than the summary from BRIO in semantics.
>
> |      | ROUGE-1 | ROUGE-2 | ROUGE-L | BERTScore |
> | ---- | ------- | ------- | ------- | --------- |
> | BRIO | 49.07   | 25.59   | 40.40   | 41.08     |
> | DSC  | 47.95   | 25.00   | 39.81   | **52.88** |
>
>
>
> **Weakness 3:**
>
> Our motivation is to select the most favorable part of the model data for training. We hope the selected data is the most favorable. Since we also consider the model to calculate the difficulty score, it is more possible to select the suitable data for the current model.

---

### Official Review · Reviewer_qyJh · 2023-08-11

**Soundness:** 2

**Excitement:**

3: Ambivalent: It has merits (e.g., it reports state-of-the-art results, the idea is nice), but there are key weaknesses (e.g., it describes incremental work), and it can significantly benefit from another round of revision. However, I won't object to accepting it if my co-reviewers champion it.

**Paper Topic And Main Contributions:**

In order to explore the impact of data selection and ordering in abstractive text summarization models, this paper firstly provides a Margin-aware List-wise Ranking(MLR) loss, which introduces a dynamic margin to improve the original loss from the BRIO. Meanwhile, get the difficulty metric mainly based on the MLR loss. Then, based on the difficulty metric value of each instance, simple and complex instances are discarded, and curriculum learning is used to training models. Experiments on the CNN/DailyMail dataset showed promising results for those improvements.

**Reasons To Accept:**

1. simple but efficient methods are provided.
2. could significantly reduce the amount of data required to train the Abstractive Text Summarization (ATS) models, and have better model performance.

**Reasons To Reject:**

Needs more experiments on other datasets to show whether the method is general.

**Reproducibility:**

5: Could easily reproduce the results.

**Reviewer Confidence:**

4: Quite sure. I tried to check the important points carefully. It's unlikely, though conceivable, that I missed something that should affect my ratings.

---

> ### Author Rebuttal · Authors · 2023-08-29
>
> Thank you very much for your response.
>
>
>
> We try to repeat our experiments on XSum from the same start point of BRIO. We initialize our model using "google/pegasus-xsum" using the selected 20% instances to train. The results are in the following table. The performance of ROUGE-2 and ROUGE-L are competitive. Besides, the BERTScore of our model beats BRIO, so the summary from our model is more similar to the reference than the summary from BRIO in semantics.
>
> |      | ROUGE-1 | ROUGE-2 | ROUGE-L | BERTScore |
> | ---- | ------- | ------- | ------- | --------- |
> | BRIO | 49.07   | 25.59   | 40.40   | 41.08     |
> | DSC  | 47.95   | 25.00   | 39.81   | **52.88** |

---

### Official Review · Reviewer_HkHS · 2023-08-14

**Soundness:** 4

**Excitement:**

3: Ambivalent: It has merits (e.g., it reports state-of-the-art results, the idea is nice), but there are key weaknesses (e.g., it describes incremental work), and it can significantly benefit from another round of revision. However, I won't object to accepting it if my co-reviewers champion it.

**Paper Topic And Main Contributions:**

This paper studies the effect of data on text summarization in the sense of order and instance selection. They propose a learning difficulty metric for data selection and curriculum learning. This score incorporates a Margin-aware List-wise Ranking loss based on the notion of contrastive learning.

The authors show through experiments on the CNN/DM dataset that these engineering techniques improve summarization performance in terms ROUGE metrics as well as BertScore.

**Questions For The Authors:**

Could you report the results of the using different percentages of the training data against ROUGE L?
Did you try to repeat these experiments on XSum? Were there issues that caused not reporting the results?



**Reasons To Accept:**

The paper touches on the interesting issue of data selection by measuring  the difficulty of each input instance. They show that this technique improves summarization performance. They authors further show that only by training using 20% of the training data the model achieves its highest performance in terms of BertScore. The paper does summarize the recent literature on abstractive summarization in an insightful way.

**Reasons To Reject:**

-The proposed method has been tested on a single dataset, CNN/DM. Therefore, the claims on the impact of data selection may not generalize to other datasets for Abstractive Summarization.
-The analysis that shows only 20% of data is needed to achieve the highest performance is only based on BertScore, however, it is very important to validate this claim also in terms of ROUGE metrics. This is due to the fact that BertScore in principle does assign high scores to semantically related sentences even if they are not exactly expressing the same stances.
-The combination of the two above issues make the claims of the paper much weaker.
-They use only a checkpoint of BRIO (which is BART-based) to initialize the model, however, if they had used also a BART checkpoint and showed similar performance it would have been helpful.



**Reproducibility:**

2: Would be hard pressed to reproduce the results. The contribution depends on data that are simply not available outside the author's institution or consortium; not enough details are provided.

**Reviewer Confidence:**

5: Positive that my evaluation is correct. I read the paper very carefully and I am very familiar with related work.

---

> ### Author Rebuttal · Authors · 2023-08-29
>
> Thanks for your efforts in reviewing our submission and for your valuable feedback on our work!
>
>
>
> We try to repeat our experiments on XSum from the same start point of BRIO. We initialize our model using "google/pegasus-xsum" using the selected 20% instances to train. The results are in the following table. The performance of ROUGE-2 and ROUGE-L are competitive. Besides, the BERTScore of our model beats BRIO, so the summary from our model is more similar to the reference than the summary from BRIO in semantics.
>
> |      | ROUGE-1 | ROUGE-2 | ROUGE-L | BERTScore |
> | ---- | ------- | ------- | ------- | --------- |
> | BRIO | 49.07   | 25.59   | 40.40   | 41.08     |
> | DSC  | 47.95   | 25.00   | 39.81   | **52.88** |
>
>
>
>
>
> The conclusion is consistent that 20% of data is needed to achieve the highest performance. Here is the result of ROUGE-L.
>
> |                    | 10%   | 20%   | 30%   | 40%   | 50%   |
> | ------------------ | ----- | ----- | ----- | ----- | ----- |
> | DSC w/o early stop | 45.02 | 45.09 | 45.0  | 44.71 | 44.69 |
> | DSC                | 45.28 | 45.31 | 45.31 | 45.31 | 45.31 |

---

### Meta-Review · Area_Chair_w8U8 · 2023-09-17

**Recommendation:** 3

**Metareview:**

The paper introduces a learning difficulty metric based on margin-aware list-wise ranking loss and curriculum learning for summarization. The incorporation of a dynamic margin to modify the original loss from the BRIO is a significant step. While the paper showcases promising results on the CNN/DM dataset with respect to both ROUGE metrics and BertScore, the validity of these findings could benefit from a broader array of tests, including other datasets and different pre-trained checkpoints. In addition, one of the reviewers is not convinced by the main conclusion, claiming optimal results in less than 20% of the instances, and believes this statement lacks sufficient rigor.

Quality:
- Limited to a single dataset (CNN/DM) for testing, raising questions about the method's generalizability.
- More validation is needed to substantiate the claim regarding the optimal amount of data for training to achieve the highest performance.

Clarity:
- While the paper has been acknowledged for its clear presentation and readability, there is a consensus among reviewers regarding the insufficient detail in explaining the core concepts, notably the difficulty metric and its utilization in curriculum learning.

Originality:
- The introduction of a dynamic margin in the loss function is somewhat novel for summarization.
- However, the curriculum learning appears more tailored for contrastive learning models, as indicated by one reviewer, requiring further justification of its applicability for abstractive text summarization.

Significance:
- The technique proposed holds promise as evidenced by the experimental results, with potential implications for improving summarization performance while reducing the amount of data necessary for training.
- Yet, the work's significance is somewhat constrained by the limited experimentation and the lack of validation across various datasets.

The paper explores an interesting research topic and integrates recent developments from other areas into text summarization effectively. However, there are significant issues with the paper: it needs clearer explanations of its methods and broader testing on various data sets to strengthen its arguments (one reviewer is concerned with over-claim). While the paper shows promising results, it requires more experiments to fully realize its potential. I am leaning towards accepting it with some reservations, but the overall soundness is not fully satisfactory (even the reviewer gave 4 for soundness indicated in the reasons for rejections many points about the claims are not backed up by enough experiments).

---

### Decision · Program_Chairs · 2023-10-07

**Decision:**

Accept-Findings

**Comment:**

The paper introduces a learning difficulty metric based on margin-aware list-wise ranking loss and curriculum learning for summarization. The incorporation of a dynamic margin to modify the original loss from the BRIO is a significant step. While the paper showcases promising results on the CNN/DM dataset with respect to both ROUGE metrics and BertScore, the validity of these findings could benefit from a broader array of tests, including other datasets and different pre-trained checkpoints. In addition, one of the reviewers is not convinced by the main conclusion, claiming optimal results in less than 20% of the instances, and believes this statement lacks sufficient rigor.

Quality:
- Limited to a single dataset (CNN/DM) for testing, raising questions about the method's generalizability.
- More validation is needed to substantiate the claim regarding the optimal amount of data for training to achieve the highest performance.

Clarity:
- While the paper has been acknowledged for its clear presentation and readability, there is a consensus among reviewers regarding the insufficient detail in explaining the core concepts, notably the difficulty metric and its utilization in curriculum learning.

Originality:
- The introduction of a dynamic margin in the loss function is somewhat novel for summarization.
- However, the curriculum learning appears more tailored for contrastive learning models, as indicated by one reviewer, requiring further justification of its applicability for abstractive text summarization.

Significance:
- The technique proposed holds promise as evidenced by the experimental results, with potential implications for improving summarization performance while reducing the amount of data necessary for training.
- Yet, the work's significance is somewhat constrained by the limited experimentation and the lack of validation across various datasets.

The paper explores an interesting research topic and integrates recent developments from other areas into text summarization effectively. However, there are significant issues with the paper: it needs clearer explanations of its methods and broader testing on various data sets to strengthen its arguments (one reviewer is concerned with over-claim). While the paper shows promising results, it requires more experiments to fully realize its potential. I am leaning towards accepting it with some reservations, but the overall soundness is not fully satisfactory (even the reviewer gave 4 for soundness indicated in the reasons for rejections many points about the claims are not backed up by enough experiments).